# ENSEMBLE AND MIXTURE-OF-EXPERTS DEEPONETS FOR OPERATOR LEARNING

## ABSTRACT

We present a novel deep operator network (DeepONet) architecture for operator learning, the ensemble DeepONet, that allows for enriching the trunk network of a single DeepONet with multiple distinct trunk networks. This trunk enrichment allows for greater expressivity and generalization capabilities over a range of operator learning problems. We also present a spatial mixture-of-experts (MoE) DeepONet trunk network architecture that utilizes a partition-of-unity (PoU) approximation to promote spatial locality and model sparsity in the operator learning problem. We first prove that both the ensemble and PoU-MoE DeepONets are universal approximators. We then demonstrate that ensemble DeepONets containing a trunk ensemble of a standard trunk, the PoU-MoE trunk, and/or a proper orthogonal decomposition (POD) trunk can achieve 2-4x lower relative $\ell_2$ errors than standard DeepONets and POD-DeepONets on both standard and challenging new operator learning problems involving partial differential equations (PDEs) in two and three dimensions. Our new PoU-MoE formulation provides a natural way to incorporate spatial locality and model sparsity into any neural network architecture, while our new ensemble DeepONet provides a powerful and general framework for incorporating basis enrichment in scientific machine learning architectures for operator learning.

## 1 INTRODUCTION

In recent years, machine learning (ML) has been applied with great success to problems in science and engineering. Notably, ML architectures have been leveraged to learn *operators*, which are function-to-function maps. In many of these applications, ML-based operators, often called *neural operators*, have been utilized to learn solution maps to partial differential equations (PDEs). This area of research, known as operator learning, has shown immense potential and practical applicability to a variety of real-world problems such as weather/climate modeling (Bora et al., 2023; Pathak et al., 2022), earthquake modeling (Haghighat et al., 2024), material science (Gupta & Brandstetter, 2022; Oommen et al., 2023), and shape optimization (Shukla et al., 2024). Some popular neural operators that have emerged are deep operator networks (DeepONets) (Lu et al., 2021), Fourier neural operators (FNOs) (Li et al., 2021), and graph neural operators (GNOs) (Li et al., 2020). DeepONets have also been extended to incorporate discretization invariance (Zhang et al., 2023), more general mappings (Jin et al., 2022), and multiscale modeling (Howard et al., 2023). In this work, we focus on the DeepONet architecture due to its ability to separate the function spaces involved in operator learning; for completeness, we discuss one possible extension to the FNO in Appendix B.

At a high level, operator learning consists of learning a map from an input function to an output function. The DeepONet architecture is an inner product between a *trunk network* that is a function of the output function domain, and a *branch network* that learns to combine elements of the trunk using transformations of the input function. In fact, one can view the trunk as a set of learned, nonlinear, data-dependent basis functions. This perspective was first leveraged to replace the trunk with a set of basis functions learned from a proper orthogonal decomposition (POD) of the training data corresponding to the output functions; the resulting POD-DeepONet achieved state-of-the-art accuracy on a variety of operator learning problems (Lu et al., 2022). More recently, this idea was further generalized by extracting a basis from the trunk as a postprocessing step (Lee & Shin, 2023); this approach proved to be highly successful in learning challenging operators (Peyvan et al., 2024).

In this work, we present the **ensemble DeepONet**, a DeepONet architecture that explicitly enables enriching a trunk network with multiple distinct trunk networks; however, this enriched/augmented

trunk uses a single branch that learns how to combine multiple trunks in such a way as to minimize the DeepONet loss function. The ensemble DeepONet essentially provides a natural framework for *basis function enrichment* of a standard (vanilla) DeepONet trunk. We also introduce a novel partition-of-unity (PoU) mixture-of-experts (MoE) trunk, **the PoU-MoE trunk**, that produces smooth blends of spatially-localized, overlapping, distinct trunks. The use of compactly-supported blending functions allows the PoU formulation to have a strong inductive bias towards spatial locality. Acknowledging that such an inductive bias is not always appropriate for learning inherently global operators, we simply introduce this PoU-MoE trunk into our ensemble DeepONet as an ensemble member alongside other global bases such as the POD trunk.

Our results show that the ensemble DeepONet, especially the POD-PoU ensemble, shows **2-4x accuracy improvements** over vanilla-DeepONets with single branches and up to **2x accuracy improvements** over the POD DeepONet (also with a single branch) in challenging 2D and 3D problems where the output function space of the operator has functions with sharp spatial gradients. In Section 4, we summarize the relative strengths of five different ensemble formulations, each carefully selected to answer a specific scientific question about the effectiveness of ensemble DeepONets. We conclude that the strength of ensemble DeepONets lie not merely in overparametrization but rather in the ability to incorporate spatially local information into the basis functions.

## 1.1 RELATED WORK

Basis enrichment has been widely used in the field of scientific computing in the extended finite element method (XFEM) (McQuien et al., 2020; Belytschko & Black, 1999; Ballard et al., 2022), modern radial basis function (RBF) methods (Flyer et al., 2016; Bayona et al., 2019; Shankar & Fogelson, 2018; Shankar et al., 2021), and others (Cai et al., 2001). In operator learning, basis enrichment (labeled "feature expansion") with trigonometric functions was leveraged to enhance accuracy in DeepONets and FNOs (Lu et al., 2022). The ensemble DeepONet generalizes these prior results by providing a natural framework to bring data-dependent, locality-aware, basis function enrichment into operator learning. PoU approximation also has a rich history in scientific computing (Melenk & Babuvska, 1996; Larsson et al., 2017; Shcherbakov & Larsson, 2016; Heryudono et al., 2016; Safdari-Vaighani et al., 2015; Shankar & Wright, 2018), and has recently found use in ML applications (Han et al., 2023; Cavoretto et al., 2021; Trask et al., 2022). In (Trask et al., 2022), which targeted (probabilistic) regression applications, the authors used trainable partition functions that were effectively black-box ML classifiers with polynomial approximation on each partition. In Han et al. (2023) (which also targeted regression), the authors used compactly-supported kernels as weight functions (like in this work), but used kernel-based regressors on each partition. Our PoU-MoE formulation generalizes both these works by using neural networks on each partition and further generalizes the technique to operator learning. In general, ensemble learning and MoE have a rich history, and we provide a more in-depth overview in Appendix A. The ensemble and PoU-MoE DeepONets introduced here extend this body of work to deterministic operator learning and PDE applications.

**Broader Impacts**: To the best of the authors' knowledge, there are no negative societal impacts of our work including potential malicious or unintended uses, environmental impact, security, or privacy concerns.

**Limitations**: Ensemble DeepONets, especially when using PoU-MoE trunks, contain 2-3x as many trainable trunk network parameters as a vanilla-DeepONet and consequently require more time to train (see Section 3.4 for runtime results and discussion); however, in future work, we plan to ameliorate this issue with a novel parallelization strategy for the PoU-MoE trunk. Further, due to limited time, we used a single branch network that outputs to $\mathbb{R}^p$ for all our results (an **unstacked branch**) rather than using $p$ branch networks that each output to $\mathbb{R}$ (a **stacked branch**) from Lu et al. (2022). This choice may result in lowered accuracy for all methods (not just ours), but certainly resulted in fewer parameters. However, our results extend straightforwardly to stacked branches also.

## 2 ENSEMBLE DEEPONETS

In this section, we first discuss the operator learning problem, then present the ensemble DeepONet architecture for learning these operators. We also present the novel PoU-MoE trunk and a modification the POD trunk from the POD-DeepONet, both for use within the ensemble DeepONet.

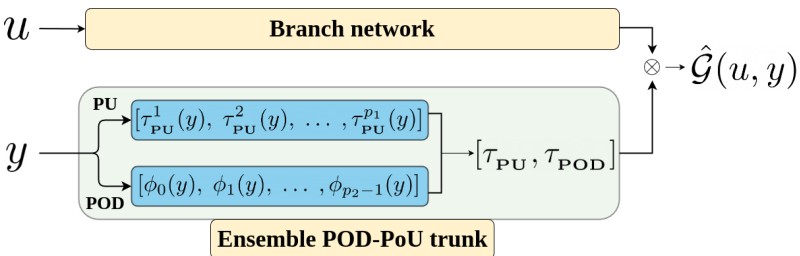

Figure 1: An ensemble DeepONet containing a POD trunk and a PoU-MoE trunk.

## 2.1 OPERATOR LEARNING WITH DEEPONETS

Let $\mathcal{U}\left(\Omega_u; \mathbb{R}^{d_u}\right)$ and $\mathcal{V}\left(\Omega_v; \mathbb{R}^{d_v}\right)$ be two separable Banach spaces of functions taking values in $\Omega_u \subset \mathbb{R}^{d_u}$ and $\Omega_v \subset \mathbb{R}^{d_v}$, respectively. Further, let $\mathcal{G} : \mathcal{U} \to \mathcal{V}$ be a general (nonlinear) operator. The operator learning problem involves approximating $\mathcal{G} : \mathcal{U} \to \mathcal{V}$ with a parametrized operator $\hat{\mathcal{G}} : \mathcal{U} \times \Theta \to \mathcal{V}$ from a finite number of function pairs $\{(u_i, v_i)\}$, $i = 1, \ldots, N$ where $u_i \in \mathcal{U}$ are typically called *input functions*, and $v_i \in \mathcal{V}$ are called *output functions, i.e.,* $v_i = \mathcal{G}(u_i)$. The parameters $\Theta$ are chosen to minimize $\|\mathcal{G} - \hat{\mathcal{G}}\|$ in some norm.

In practice, the problem must be discretized. First, one puts samples the input and output functions at a finite set of function sample locations $X \in \Omega_u$ and $Y \in \Omega_v$, respectively; also let $N_x = |X|$ and $N_y = |Y|$. One then requires that $\|v_i(y) - \hat{\mathcal{G}}(u_i)(y)\|_2^2$ is minimized over $(u_i, v_i)$, $i = 1, \ldots, N$, where $u_i$ are sampled at $x \in X$ and $v_i$ at $y \in Y$. The vanilla-DeepONet is one particular parametrization of $\hat{\mathcal{G}}(u)(y)$ as $\hat{\mathcal{G}}(u)(y) = \langle \boldsymbol{\tau}(y), \boldsymbol{\beta}(u) \rangle + b_0$ where $\langle , \rangle$ is the $p$-dimensional inner product, $\boldsymbol{\beta} : \mathbb{R}^{N_x} \times \Theta_{\boldsymbol{\beta}} \to \mathbb{R}^p$ is the *branch* (neural) network, $\boldsymbol{\tau} : \mathbb{R}^{d_v} \times \Theta_{\boldsymbol{\tau}} \to \mathbb{R}^p$ is the trunk network, and $b_0$ is a trainable bias parameter; $p$ is a hyperparameter that partly controls the expressivity of $\hat{\mathcal{G}}(u)(y)$. $\Theta_{\boldsymbol{\beta}}$ and $\Theta_{\boldsymbol{\tau}}$ are the trainable parameters in the branch and trunk, respectively.

## 2.2 MATHEMATICAL FORMULATION

We now present the new ensemble DeepONet formulation; an example is illustrated in Figure 1. Without loss of generality, assume that we are given three distinct trunk networks $\boldsymbol{\tau}_1(y; \theta_{\boldsymbol{\tau}_1}), \boldsymbol{\tau}_2(y; \theta_{\boldsymbol{\tau}_2})$, and $\boldsymbol{\tau}_3(y; \theta_{\boldsymbol{\tau}_3})$, where $y$ corresponds to the domain of the output function $v(y)$. Assume further that $\boldsymbol{\tau}_j : \mathbb{R}^d \times \Theta_{\boldsymbol{\tau}_j} \to \mathbb{R}^{p_j}$, $j = 1, 2, 3$. Then, given a single branch network $\hat{\boldsymbol{\beta}}(u; \theta_b)$, the **ensemble DeepONet** is given in vector form by:

$$\hat{\mathcal{G}}(u, y) = \left\langle [\boldsymbol{\tau}_1(y; \theta_{\boldsymbol{\tau}_1}), \boldsymbol{\tau}_2(y; \theta_{\boldsymbol{\tau}_2}), \boldsymbol{\tau}_3(y; \theta_{\boldsymbol{\tau}_3})], \hat{\boldsymbol{\beta}}(u; \theta_b) \right\rangle + b_0 = \left\langle \hat{\boldsymbol{\tau}}, \hat{\boldsymbol{\beta}}(u; \theta_b) \right\rangle + b_0. \quad (1)$$

Here, $\hat{\boldsymbol{\tau}} : \mathbb{R}^{d_v} \times \Theta_{\boldsymbol{\tau}_1} \times \Theta_{\boldsymbol{\tau}_2} \times \Theta_{\boldsymbol{\tau}_3} \to \mathbb{R}^{p_1 + p_2 + p_3}$ is the *ensemble trunk*. Clearly, the individual trunks simply "stack" column-wise to form the ensemble trunk $\hat{\boldsymbol{\tau}}$; in Appendix C, we discuss other suboptimal attempts to form an ensemble trunk. The ensemble trunk now consists of $p_1 + p_2 + p_3$ (potentially trainable) basis functions, necessitating that the branch $\hat{\boldsymbol{\beta}} : \mathbb{R}^{N_x} \times \Theta_{\hat{\boldsymbol{\beta}}} \to \mathbb{R}^{p_1 + p_2 + p_3}$.

**A universal approximation theorem**

**Theorem 1.** *Let* $\mathcal{G} : \mathcal{U} \to \mathcal{V}$ *be a continuous operator. Define* $\hat{\mathcal{G}}$ *as* $\hat{\mathcal{G}}(u, y) = \left\langle \hat{\boldsymbol{\tau}}(y; \theta_{\boldsymbol{\tau}_1}; \theta_{\boldsymbol{\tau}_2}; \theta_{\boldsymbol{\tau}_3}), \hat{\boldsymbol{\beta}}(u; \theta_b) \right\rangle + b_0$, *where* $\hat{\boldsymbol{\beta}} : \mathbb{R}^{N_x} \times \Theta_{\hat{\boldsymbol{\beta}}} \to \mathbb{R}^{p_1 + p_2 + p_3}$ *is a branch network embedding the input function* $u$, $b_0$ *is the bias, and* $\hat{\boldsymbol{\tau}} : \mathbb{R}^{d_v} \times \Theta_{\hat{\boldsymbol{\tau}}_1} \times \Theta_{\hat{\boldsymbol{\tau}}_2} \times \Theta_{\hat{\boldsymbol{\tau}}_3} \to \mathbb{R}^{p_1 + p_2 + p_3}$ *is an* ensemble *trunk network. Then* $\hat{\mathcal{G}}$ *can approximate* $\mathcal{G}$ *globally to any desired accuracy, i.e.,*

$$\mathcal{G}(u)(y) - \hat{\mathcal{G}}(u)(y)\|_{\mathcal{V}} \le \epsilon, \quad (2)$$

*where* $\epsilon > 0$ *can be made arbitrarily small.*

*Proof.* This automatically follows from the (generalized) universal approximation theorem (Lu et al., 2021) which holds for arbitrary branches and trunks. $\qquad\square$

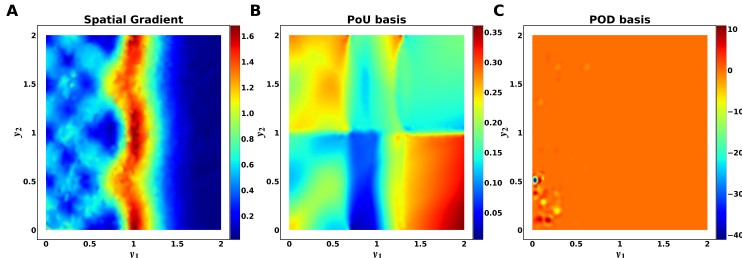

Figure 2: Enriched bases on the 2D **reaction-diffusion** problem 3.2. The solutions exhibit sharp gradients (left); the PoU-MoE trunk has learned spatially-localized basis functions (middle); the POD trunk has learned a global basis function (right).

### 2.2.1 THE POU-MOE TRUNK

We now present the PoU-MoE trunk architecture, which leverages partition-of-unity approximation. We begin by partitioning $\Omega_v$ into $P$ overlapping circular/spherical patches $\Omega_k$, $k = 1, \ldots, P$, with each patch having its own radius $\rho_k$ and containing a set of sample locations $Y_k$; of course, $\bigcup_{k=1}^{P} Y_k = Y$. The key idea behind the PoU-MoE trunk is to employ a separate trunk network on each patch $\Omega_k$ and then blend (and train) these trunks appropriately to yield a **single** trunk network on $\Omega$. Each $\boldsymbol{\tau}_k$ is trained at data on $Y_k$, but may also be influenced by spatial neighbors. The PoU-MoE trunk $\boldsymbol{\tau}_{\text{PU}}(x)$ is given as follows:

$$\boldsymbol{\tau}_{\text{PU}}(y; \theta_{\boldsymbol{\tau}_{\text{PU}}}) = \sum_{k=1}^{P} w_k(y)\boldsymbol{\tau}_k(y; \theta_{\boldsymbol{\tau}_k}), \tag{3}$$

where $\theta_{\boldsymbol{\tau}_k}$, $k = 1, \ldots, P$ are the trainable parameters for each trunk. In this work, we choose the weight functions $w_k$ to be (scaled and shifted) compactly-supported, positive-definite kernels $\psi_k : \mathbb{R}^d \times \mathbb{R}^d \to \mathbb{R}$ that are $\mathbb{C}^2(\mathbb{R}^d)$. More specifically, on the patch $\Omega_k$, we select $\psi_k$ to be the $\mathbb{C}^2(\mathbb{R}^3)$ Wendland kernel (Wendland, 1995; 2005; Fasshauer, 2007; Fasshauer & McCourt, 2015), which is a *radial* kernel given by

$$\psi_k(y, y^c) = \psi_k\left(\frac{\|y - y_k^c\|}{\rho_k}\right) = \psi_k(r) = \begin{cases} (1 - r)^4(4r + 1), & \text{if } r \leq 1 \\ 0, & \text{if } r > 1 \end{cases}, \tag{4}$$

where $y_k^c$ is the center of the $k$-th patch. The weight functions are then given by

$$w_k(y) = \frac{\psi_k(y)}{\sum_j \psi_j(y)}, \ k, j = 1, \ldots, P, \tag{5}$$

which automatically satisfy $\sum_k w_k(y) = 1$. Each trunk $\boldsymbol{\tau}_k$ can be viewed as an "expert" on its own patch $\Omega_k$, thus leading to a *spatial* MoE formulation via the PoU formalism. Both training and evaluation of $\boldsymbol{\tau}_{\text{PU}}$ can proceed locally in that each location $y$ lies in only a few patches; our implementation leverages this fact for efficiency. Further, since the weight functions $w_k(y)$ are each compactly-supported on their own patches $\Omega_k$, $\boldsymbol{\tau}_{\text{PU}}$ can be viewed as *sparse* in its constituent spatial experts $\boldsymbol{\tau}_k$. Nevertheless, by ensuring that neighboring patches overlap sufficiently, we ensure that $\boldsymbol{\tau}_{\text{PU}}$ still constitutes a global set of basis functions. For simplicity, we use the same $p$ value within each local trunk $\boldsymbol{\tau}_k$. Figure 2 (middle) shows one of the learned PoU-MoE basis functions in the POD-PoU ensemble; the learned basis function exhibits strong spatial locality corresponding to partitions. In Appendix G, we present more evidence for this spatial localization in the PoU-MoE basis functions.

**Partitioning:** We placed the patch centers in a bounding box around $\Omega$, place a Cartesian grid in that box, then simply select $P$ of the grid points to use as centers. In this case, the uniform radius $\rho$ is determined as (Larsson et al., 2017) $\rho = (1 + \delta)0.5H\sqrt{d}$ where $\delta$ is a free parameter to describe the overlap between patches and $H$ is the side length of the bounding box. However, as a demonstration, we also used variable radii $\rho_k$ in Section 3.1. In this work, we placed patches by using spatial gradients of a vanilla-DeepONet as our guidance, attempting to balance covering the whole domain with resolving these gradients; see Appendix E.3 for a more in-depth discussion on partitioning strategies.

**A universal approximation theorem**

**Theorem 2.** *Let* $\mathcal{G} : \mathcal{U} \to \mathcal{V}$ *be a continuous operator. Define* $\mathcal{G}^\dagger$ *as* $\mathcal{G}^\dagger(u)(y) = \left\langle \boldsymbol{\beta}(u; \theta_b), \sum_{j=1}^{P} w_j(y)\boldsymbol{\tau}_j(y; \theta_{\boldsymbol{\tau}_j}) \right\rangle + b_0$, *where* $\boldsymbol{\beta} : \mathbb{R}^{N_x} \times \Theta_{\boldsymbol{\beta}} \to \mathbb{R}^p$ *is a branch network embedding the input function* $u$, $\boldsymbol{\tau}_j : \mathbb{R}^{d_v} \times \Theta_{\boldsymbol{\tau}_j} \to \mathbb{R}^p$ *are trunk networks,* $b_0$ *is a bias, and* $w_j : \mathbb{R}^{d_v} \to \mathbb{R}$ *are compactly-supported, positive-definite weight functions that satisfy the partition of unity condition* $\sum_j w_j(y) = 1, j = 1, \ldots, P$. *Then* $\mathcal{G}^\dagger$ *can approximate* $\mathcal{G}$ *globally to any desired accuracy,* i.e.,

$$\mathcal{G}(u)(y) - \mathcal{G}^\dagger(u)(y)\|_{\mathcal{V}} \leq \epsilon, \tag{6}$$

*where* $\epsilon > 0$ *can be made arbitrarily small.*

*Proof.* See Appendix D for the proof. The high level idea is to use the fact that the (generalized) universal approximation theorem (Chen & Chen, 1995; Lu et al., 2021) already holds for each local trunk on a patch, then use the partition of unity property to effectively blend that result over all patches to obtain a global estimate. $\square$

### 2.2.2 THE POD TRUNK

The POD trunk is a modified version of the trunk used in the POD-DeepONet (Chatterjee, 2000) of the output function data. First, we remind the reader of the POD procedure. Recalling that $\{v_i(y)\}_{i=1}^{N}$ are the output functions, first define the matrix $V_{ij} = \frac{1}{\sigma_i}(v_i(y_j) - \mu_i)$, where $\mu_i$ is the spatial mean of the $i$-th function and $\sigma_i$ is its spatial standard deviation. Define the matrix $T = \frac{1}{N}VV^T$, and let $\Phi$ be the matrix of eigenvectors of $T$ ordered from the smallest eigenvalue to the largest. Then, the POD-DeepONet involves selecting the first $p$ columns of $\Phi$ to be the trunk of a DeepONet so that $G_{\text{POD}}(u, y) = \sum_{i=1}^{p} \boldsymbol{\beta}_i(u)\phi_i(y) + \phi_0(y)$, where $\phi_0(y)$ is the mean function of $v(y)$ computed from the training dataset, and $\phi_i(y)$ are the columns of $\Phi$ as explained above. In this work, we use a POD trunk that includes the mean function $\phi_0$ in the set of basis functions. We label this the "Modified-POD" trunk in our experiments; this "Modified-POD" trunk $\boldsymbol{\tau}_{\text{POD}}$ is given by

$$\boldsymbol{\tau}_{\text{POD}}(y) = \begin{bmatrix} \phi_0(y) & \phi_1(y) & \ldots & \phi_{p-1}(y) \end{bmatrix}, \tag{7}$$

Consistent with the POD-DeepONet philosophy, no activation function is needed and the POD trunk has no trainable parameters. Figure 2 (right) shows one of the learned POD basis functions in the POD-PoU ensemble.

### 2.2.3 OTHER NEURAL OPERATORS

While we restricted our attention to DeepONets in this work, the ensemble idea naturally extends to other neural operator architectures. In Appendix B, we briefly discuss our ideas on creating ensembles of global and local basis functions within the FNO.

## 3 RESULTS

We present results of our comparison of the new ensemble DeepONet (with and without a PoU-MoE trunk) against vanilla and POD DeepONets. We considered different ensemble combinations of the vanilla, POD, and PoU-MoE trunks. Each of the following ensembles attempted to address a specific scientific question:

1. **Vanilla-POD**: Does adding POD modes to a vanilla trunk enhance expressivity over using either trunk in isolation?
2. **Vanilla-PoU**: Does spatial locality introduced by the PoU-MoE trunk aid a DeepONet?
3. **POD-PoU**: Does having both POD global modes and PoU-MoE local expertise enhance expressivity over simply using a vanilla trunk?
4. **Vanilla-POD-PoU**: If the answer above is affirmative, then does adding a vanilla trunk (representing extra trainable parameters) to a POD-PoU ensemble help further enhance expressivity?
5. **$(P+1)$-Vanilla**: Is spatial localization truly important or is simple overparametrization all that is needed? We use $P+1$ vanilla trunks in this model, where $P$ is the number of PoU-MoE patches. This ensemble thus contains as many trunks as the vanilla-PoU or POD-PoU ensembles, but all basis functions are purely *global* in this setting.

Table 1: Relative $l_2$ errors (as percentage) on the test dataset for the 2D **Darcy flow**, **cavity flow**, and **reaction-diffusion**, and the 3D **reaction-diffusion** problems. RD stands for reaction-diffusion.

|  | Darcy flow | Cavity flow | 2D RD | 3D RD |
|---|---|---|---|---|
| Vanilla | $0.857 \pm 0.08$ | $5.53 \pm 1.05$ | $0.144 \pm 0.01$ | $0.127 \pm 0.03$ |
| POD | $0.297 \pm 0.01$ | $7.94 \pm 2e-5$ | $5.06 \pm 8e-7$ | $9.40 \pm 8$ |
| Modified-POD | $0.300 \pm 0.04$ | $7.93 \pm 2e-5$ | $0.131 \pm 4e-5$ | $0.155 \pm 4e-5$ |
| (Vanilla, POD) | $0.227 \pm 0.03$ | $0.310 \pm 0.03$ | $0.0751 \pm 4e-5$ | $5.24 \pm 10.4$ |
| $(P+1)$-Vanilla | $1.19 \pm 0.06$ | $2.17 \pm 0.3$ | $0.0644 \pm 0.02$ | $5.25 \pm 10.3$ |
| (Vanilla, PoU) | $0.976 \pm 0.03$ | $1.06 \pm 0.05$ | $0.0946 \pm 0.03$ | $5.25 \pm 10.3$ |
| (POD, PoU) | $0.204 \pm 0.02$ | $\mathbf{0.204 \pm 0.01}$ | $\mathbf{0.0539 \pm 4e-5}$ | $\mathbf{0.0576 \pm 0.05}$ |
| (Vanilla, POD, PoU) | $\mathbf{0.187 \pm 0.02}$ | $0.229 \pm 0.01$ | $0.0666 \pm 8e-5$ | $5.22 \pm 10.4$ |

The answers to these questions are shown in Table 4 and summarized in Section 4. In a nutshell, spatial localization is indeed important, as is using a mix of global and localized basis functions; simple overparametrization is insufficient to attain state-of-the-art accuracy. We now describe our experimental setup, and both the standard and novel benchmark test results that led us to this conclusion.

**Important DeepONet details.** In all cases, for parsimony in the number of training parameters, we used a single branch (the unstacked DeepONet) that outputs to $\mathbb{R}^p$ rather than $p$ branches. We found that output normalization did not help significantly in this case. We scaled all our POD architecture outputs by $\frac{1}{p}$ (standalone or in ensembles), as advocated in Lu et al. (2022).

**Experiment design.** In the remainder of this section, we establish the performance of ensemble DeepONets on benchmarks such as a 2D lid-driven cavity flow problem (Section 3.1) and a 2D Darcy flow problem on a triangle (Appendix F.1), both common in the literature (Lu et al., 2022; Batlle et al., 2024). However, we also wished to develop challenging new spacetime PDE benchmarks where the PDE solutions (output functions) possessed steep gradients, while the input functions were well-behaved. To this end, we present results for both a 2D reaction-diffusion problem (Section 3.2) and a 3D reaction-diffusion problem with sharply (spatially) varying diffusion coefficients (Section 3.3). In both cases, we constructed **spatially discontinuous** reaction terms that resulted in PDE solutions (output functions) with steep gradients. Such PDE solutions abound in scientific applications. **We note at the outset that the ensemble DeepONet with the PoU-MoE trunk performed best when the solutions had steep spatial gradients. Results on the Darcy problem show that the ensemble approaches tested here were not as effective on that problem**.

**Error calculations.** For all problems, we compared the vanilla- and POD-DeepONets with the five different ensemble architectures described at the top of Section 3. We also compared these ensembles against a DeepONet with the modified POD trunk from Section 2.2.2 (labeled **Modified-POD**). For all experiments, we first computed the relative $l_2$ error for each test function, $e_{\ell_2} = \frac{\|\tilde{u}-u\|_2}{\|u\|_2}$ where $\underline{u}$ was the true solution vector and $\underline{\tilde{u}}$ was the DeepONet prediction vector; we then computed the mean over those relative $\ell_2$ errors. For vector-valued functions, we first computed pointwise magnitudes of the vectors, then repeated the same process. We also report a squared error (MSE) between the DeepONet prediction and the true solution averaged over $N$ *functions* $e_{\mathrm{mse}}(y) = \frac{1}{N} \left( \tilde{u}(y) - u(y) \right)^2$.

**Notation.** In the following text, we denote the space and time domains with $\Omega$ and $T$ respectively; the spatial domain boundary is denoted by $\partial\Omega$. A single spatial point is denoted by $y$, which can either be a point $(y_1, y_2)$ in $\mathbb{R}^2$ or a point $(y_1, y_2, y_3)$ in $\mathbb{R}^3$.

**Setup.** We trained all models for 150,000 epochs on an NVIDIA GTX 4080 GPU. All results were calculated over five random seeds. We annealed the learning rates with an inverse-time decay schedule. We used the Adam optimizer (Kingma & Ba, 2017) for training on the Darcy flow and the cavity flow problems, and the AdamW optimizer (Loshchilov & Hutter, 2018) on the 2D and 3D reaction-diffusion problems. Other DeepONet hyperparameters and the network architectures are listed in Appendix E.

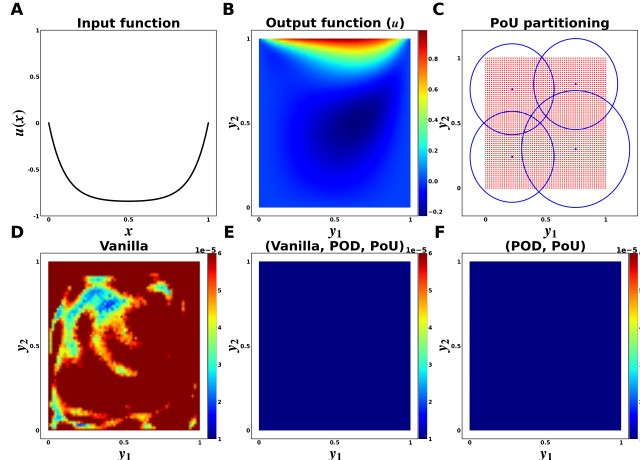

Figure 3: The 2D lid-driven **cavity flow** problem. We show in (**A**) an example input function; in (**B**) an example output function component; in (**C**) the four patches used for the PoU-MoE trunk; in (**D**), (**E**), and (**F**) the spatial mean squared error (MSE) for the vanilla, ensemble vanilla-POD-PoU, and ensemble POD-PoU DeepONets respectively.

### 3.1    2D LID-DRIVEN CAVITY FLOW

The 2D lid-driven cavity flow problem involves solving for fluid flow in a container whose lid moves tangentially along the top boundary. This can be described by the incompressible Navier-Stokes equations (with boundary conditions),

$$\frac{\partial \mathbf{u}}{\partial t} + (\mathbf{u} \cdot \nabla)\mathbf{u} = -\nabla \mathbf{p} + \nu \Delta \mathbf{u}, \ \ \nabla \cdot \mathbf{u} = 0, \ y \in \Omega, \ t \in T, \tag{8}$$

$$\mathbf{u} = \mathbf{u}_b, \tag{9}$$

where $\mathbf{u} = (u(y), v(y))$ is the velocity field, $p$ is the pressure field, $\nu$ is the kinematic viscosity, and $\mathbf{u}_b = (u_b, v_b)$ is the Dirichlet boundary condition. We focused on the steady state problem and used the dataset specified in Lu et al. (2022, Section 5.7, Case A). We set $\Omega = [0, 1]^2$ and learned the operator $\mathcal{G} : \mathbf{u}_b \to \mathbf{u}$. The steady state boundary condition is defined as,

$$u_b = U \left( 1 - \frac{\cosh\left(r(x - \frac{1}{2})\right)}{\cosh\left(\frac{r}{2}\right)} \right), \ \ v_b = 0, \tag{10}$$

where $r = 10$. The other boundary velocities were set to zero. As described in Lu et al. (2022), the equations were then solved using a lattice Boltzmann method (LBM) to generate 100 training and 10 test input and output function pairs. All function pairs were generated over a range of Reynolds numbers in the range $[100, 2080]$ (with $U$ and $\nu$ chosen appropriately), with no overlap between the training and test dataset. Figure 3 shows the four patches used to partition the domain.

We report the relative $\ell_2$ errors (as percentage) on the test dataset in Table 1. The vanilla-, modified POD-, and POD-DeepONets had the highest errors (in increasing order). The POD-PoU ensemble was the most accurate model by about an order of magnitude over the vanilla-DeepONet, and almost two orders of magnitude over the POD variants. While all ensembles outperformed the standalone DeepONets, the ensembles possessing POD modes appeared to do best in general. Further, adding a PoU-MoE trunk to the ensemble seemed to aid accuracy in general, but especially when POD modes were present. The spatial MSE figures in Figure 3 reflect the same trends.

### 3.2    A 2D REACTION-DIFFUSION PROBLEM

Next, we present experimental results on a 2D reaction-diffusion problem. This equation governs the behavior of a chemical whose concentration is $c(y, t)$, and is given (along with boundary conditions) below:

$$\frac{\partial c}{\partial t} = k_{\text{on}} \left(R - c\right) c_{\text{amb}} - k_{\text{off}} \, c + \nu \Delta c, \ y \in \Omega, \ t \in T, \tag{11}$$

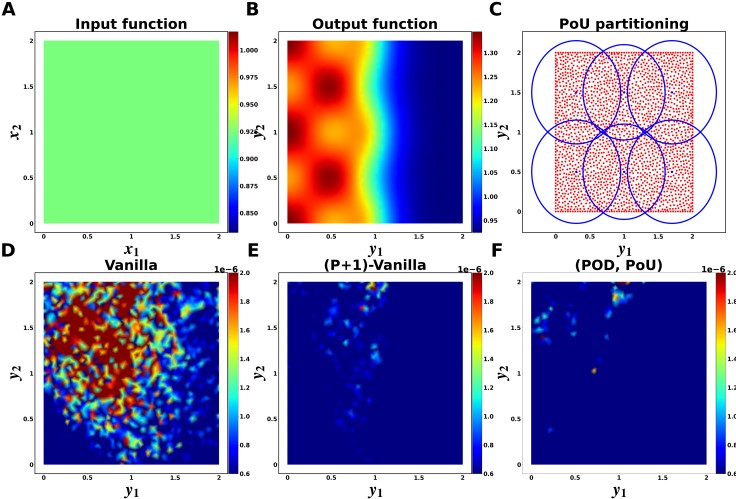

Figure 4: The 2D **reaction-diffusion** problem. We show in (**A**) an example input function; in (**B**) an example output function; in (**C**) the six patches used for the PoU-MoE trunk; in (**D**), (**E**), and (**F**) the spatial mean squared error (MSE) for the vanilla, ensemble $(P + 1)$-vanilla, and ensemble POD-PoU DeepONets respectively.

with the boundary condition $\nu \frac{\partial c}{\partial n} = 0$ on $\partial \Omega$. The first r.h.s term is a binding reaction term modulated by $k_{\text{on}}$ and the second term an unbinding term modulated by $k_{\text{off}}$. $c_{\text{amb}}(y, t) = 1 + \cos(2\pi y_1)\cos(2\pi y_2))\exp(-\pi t)$ is a background source of chemical available for reaction, $\nu = 0.1$ is the diffusion coefficient, $R = 2$ is a throttling term, and $n(y)$ is the unit outward normal vector on the boundary. In our experiments, we used $\Omega = [0, 2]^2$ and $T = [0, 0.5]$. We set the initial condition as a spatial constant $c(y, 0) \sim \mathcal{U}(0, 1)$. More importantly, $k_{\text{on}}$ and $k_{\text{off}}$ are discontinuous and given by

$$k_{\text{on}} = \begin{cases} 2, & y_1 \leq 1.0, \\ 0, & \text{otherwise} \end{cases}, \quad k_{\text{off}} = \begin{cases} 0.2, & y_1 \leq 1.0, \\ 0, & \text{otherwise} \end{cases}, \tag{12}$$

where $y_1$ is the horizontal direction. This discontinuity induces a sharp solution gradient at $y_1 = 1.0$ (see Figure 4 (B)). Our goal was to learn the solution operator $G : c(y, 0) \rightarrow c(y, 0.5)$. We solved the PDE numerically at $N_y = 2207$ collocation points using a fourth-order accurate RBF-FD method (Shankar & Fogelson, 2018; Shankar et al., 2021); using this solver, we generated 1000 training and 200 test input and output function pairs. We sampled the random spatially-constant input on a regular spatial grid for the branch input. We used six patches for the PoU trunks as shown in Figure 4.

The third column of Table 1 shows that the POD-PoU ensemble achieved the lowest error, with an error reduction of almost 3x over the standalone DeepONets. The $(P + 1)$-vanilla ensemble also performed reasonably well, with a greater than 2x error reduction over the same; this indicates that overparametrization indeed helped on this test case. However, the relatively higher errors of the vanilla-PoU ensemble (compared to the best results) indicate that POD modes are possibly vital to fully realizing the benefits of the PoU-MoE trunk. Once again, the spatial MSE plots in Figure 4 corroborate the relative errors.

### 3.3 3D Reaction-Variable-Coefficient-Diffusion

Finally, we present results on a 3D reaction-diffusion problem with *variable-coefficient diffusion*. We used a similar setup to the 2D case but significantly also allow the diffusion coefficient to vary spatially via a function $K(y)$, $y \in \mathbb{R}^3$. The PDE and boundary conditions are given by

$$\frac{\partial c}{\partial t} = k_{\text{on}}(R - c) c_{\text{amb}} - k_{\text{off}} c + \nabla \cdot (K(y)\nabla c), \ y \in \Omega, \ t \in T, \tag{13}$$

with $K(y)\frac{\partial c}{\partial n} = 0$ on $\partial \Omega$. Here, $\Omega$ was the unit ball, *i.e.*, the interior of the unit sphere $\mathbb{S}^2$, and $T = [0, 0.5]$. We set the $k_{\text{on}}$ and $k_{\text{off}}$ coefficients to the same values as in 2D in $y_1 \leq 0$, and to zero in

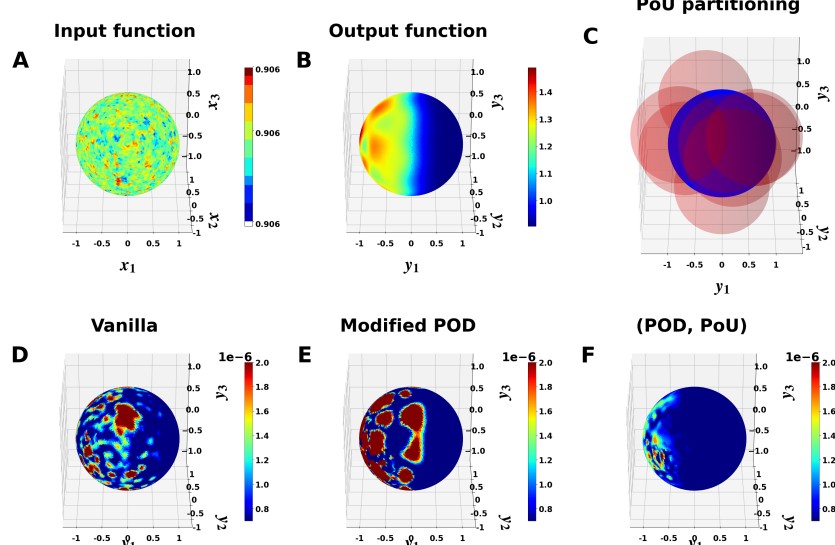

Figure 5: The 3D **reaction-diffusion** problem. We show in (**A**) an example input function; in (**B**) an example output function; in (**C**) the eight patches used for the PoU-MoE trunk; in (**D**), (**E**), and (**F**) the spatial mean squared error (MSE) for the vanilla, modified POD, and ensemble POD-PoU DeepONets respectively.

the $y_1 > 0$ half of the domain. We set $c_{\mathrm{amb}} = (1 + \cos(2\pi y_1)\cos(2\pi y_2)\sin(2\pi y_3))e^{(-\pi t)}$. All other model parameters were kept the same. $K(y)$ was chosen to have steep gradients, here defined as

$$K(y) = B + \frac{C}{\tanh(A)}\left((A-3)\tanh(8x-5) - (A-15)\tanh(8x+5) + A\tanh(A)\right), \quad (14)$$

where $A = 9$, $B = 0.0215$, and $C = 0.005$. Once again, we learned the operator $\mathcal{G} : c(y, 0) \rightarrow c(y, 0.5)$. We again used the same RBF-FD solver to generate 1000 training and 200 test input/output function pairs (albeit at 4325 collocation points in 3D). We used eight spatial patches for the PoU trunks as shown in Figure 5. The last column in Table 1 shows that most of the ensemble DeepONets did poorly, as did the POD-DeepONet. However, the POD-PoU ensemble achieved almost a 2x reduction in error over the vanilla-DeepONet.

### 3.4 RUNTIME COMPARISON

The ensemble DeepONet architectures all have more trainable parameters than the vanilla and POD DeepOnets. This leads to higher training and inference times. We report the average time per training epoch and inference time on the test dataset in Tables 2 and 3 respectively. The training times were larger in ensemble DeepONets with more trunk networks, considerably so when the PoU-MoE trunks were used (an order of magnitude increase in training time on the 3D reaction-diffusion problem). The inference times showed a similar trend, although much less pronounced (only half an order of magnitude slowdown in the 3D problem). These slowdowns are because our current PoU-MoE implementation contains a serial loop over the patches in the forward pass, leading to slower back-propagation over its parameters. In future work, we plan to address this with a novel parallelization strategy; we believe this will speed up the ensemble architectures with the PoU-MoE trunk considerably. It is also important to note that despite this increased cost, Table 1 shows that the POD-PoU ensemble is more than 2x as accurate as the vanilla-DeepONet; the POD-DeepONet (and other ensembles) have errors that are **two orders of magnitude worse!**

### 4 CONCLUSIONS AND FUTURE WORK

We presented the ensemble DeepONet, a method of enriching a DeepONet trunk with arbitrary trunks. We also developed the PoU-MoE trunk to aid in spatial locality. Our results demonstrated significant accuracy improvements over standalone DeepONets on several challenging operator learning problems, including a particularly challenging 3D problem in the unit ball. One of the

Table 2: Average time per training epoch in seconds. RD stands for reaction-diffusion.

|  | Darcy flow | Cavity flow | 2D RD | 3D RD |
|---|---|---|---|---|
| Vanilla | $8.93e-4$ | $3.99e-4$ | $2.97e-4$ | $2.10e-4$ |
| POD | $5.19e-4$ | $2.46e-4$ | $2.06e-4$ | $1.22e-4$ |
| Modified-POD | $6.86e-4$ | $2.49e-4$ | $2.08e-4$ | $1.22e-4$ |
| (Vanilla, POD) | $9.80e-4$ | $3.92e-4$ | $3.03e-4$ | $2.32e-4$ |
| $(P+1)$-Vanilla | $1.10e-3$ | $8.51e-4$ | $7.27e-4$ | $9.45e-4$ |
| Vanilla-PoU | $8.67e-4$ | $9.52e-4$ | $1.03e-3$ | $1.39e-3$ |
| POD-PoU | $6.74e-4$ | $8.21e-4$ | $9.24e-4$ | $1.28e-3$ |
| Vanilla-POD-PoU | $8.55e-4$ | $9.48e-4$ | $1.05e-3$ | $1.43e-3$ |

Table 3: Inference time on the test dataset in seconds. RD stands for reaction-diffusion.

|  | Darcy flow | Cavity flow | 2D RD | 3D RD |
|---|---|---|---|---|
| Vanilla | $1.66e-4$ | $1.39e-4$ | $1.32e-4$ | $7.20e-5$ |
| POD | $1.57e-4$ | $1.12e-4$ | $1.12e-4$ | $6.42e-5$ |
| Modified-POD | $1.34e-4$ | $1.08e-4$ | $9.94e-5$ | $6.62e-5$ |
| (Vanilla, POD) | $1.69e-4$ | $1.33e-4$ | $1.20e-4$ | $7.76e-5$ |
| $(P+1)$-Vanilla | $2.08e-4$ | $2.12e-4$ | $1.71e-4$ | $1.48e-4$ |
| Vanilla-PoU | $1.91e-4$ | $2.42e-4$ | $2.21e-4$ | $2.37e-4$ |
| POD-PoU | $1.63e-4$ | $1.94e-4$ | $1.96e-4$ | $2.30e-4$ |
| Vanilla-POD-PoU | $2.00e-4$ | $2.18e-4$ | $2.28e-4$ | $2.41e-4$ |

Table 4: Effectiveness of different trunk choices. The yes/no refers to whether the strategy beats a vanilla-DeepONet. The bolded results are the best strategy for each experiment. RD stands for reaction-diffusion.

| Trunk Choices | Darcy flow | Cavity flow | 2D RD | 3D RD |
|---|---|---|---|---|
| Only POD global modes | Yes | No | No | No |
| Only modified POD global modes | Yes | No | No | No |
| Adding POD global modes | Yes | Yes | Yes | No |
| Adding spatial locality | No | Yes | Yes | No |
| Only POD global modes + spatial locality | Yes | **Yes** | **Yes** | **Yes** |
| Only POD global modes + spatial locality + mild overparametrization | **Yes** | Yes | Yes | No |
| Adding excessive overparametrization | No | Yes | Yes | No |

goals of this work was to provide insight into choices for ensemble trunk members. Thus, we considered different combinations of three very specific choices: a vanilla-DeepONet trunk (vanilla trunk), the POD trunk, and the new PoU-MoE trunk. Our results (summarized in Table 4) make clear that while different ensemble strategies beat the vanilla-DeepONet in different circumstances, only the POD-PoU ensemble consistently beats the vanilla-DeepONet across all problems. Simple overparametrization ($(P+1)$-Vanilla DeepONet) is not enough and sometimes deteriorates accuracy; a judicial combination of local and global basis functions is vital. Further, adding the PoU-MoE trunk aids expressivity in every problem that involves steep spatial gradients in either the input or output functions. Finally, it appears that the full benefits of the PoU-MoE trunk are mainly achieved when the POD trunk is also used in the ensemble.

Given the generality of our work, there are numerous possible extensions along the lines of problem-dependent choices for the ensemble members. The PoU-MoE trunk merits further investigation. It is plausible that adding adaptivity to the PoU weight functions could improve its accuracy further, as could a spatially hierarchical formulation. Our work also paves the way for the use of other non-neural network basis functions within the ensemble DeepONet.

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

## A    Ensemble Learning and Mixture-of-Experts (MoE)

The key idea behind ensemble learning is to combine a diverse set of learnable features from individual models into a single model (Polikar, 2012; Dong et al., 2020; Zhou, 2021; Dasarathy & Sheela, 1979). This technique has been used for both supervised and unsupervised feature selection in a variety of applications (Saeys et al., 2008; Li et al., 2012; Elghazel & Aussem, 2015; Abeel et al., 2010; Van Landeghem et al., 2010; Awada et al., 2012; Dittman et al., 2012; Piao et al., 2012). Guan et al. (2014) draws an important distinction between using ensemble learning for feature selection and using feature selection for ensemble learning (where the former category is known to overcome the problem of local minima in machine learning). It is generally known that these methods are more stable than single base learners (Tuv, 2006; Guan et al., 2014). While ensemble methods have traditionally been studied with a statistical perspective (Krogh & Sollich, 1997; Chipman et al., 2006), we focus more on its feature selection capability, i.e. our work falls in the category of ensemble learning for feature selection. We use ensemble learning specifically to aggregate the global and local spatial features learned by the vanilla, POD, and PoU-MoE trunks into a single ensemble trunk.

MoE, first introduced in (Jacobs et al., 1991; Jordan & Jacobs, 1993), is a method in which an "expert" model focuses on learning from a subset of the training dataset. These models can be support vector machines (Lima et al., 2007; Lima et al., 2009; Collobert et al., 2001), Gaussian Processes (Ng & Deisenroth, 2014; Yuan & Neubauer, 2008; Gadd et al., 2020), and neural networks. Similar to ensemble learning, the MoE idea has also proven to be very successful in diverse ML applications (Masoudnia & Ebrahimpour, 2014; Yuksel et al., 2012; Chen et al., 2022). Most recently, MoE has also been used in physics-informed learning; Chalapathi et al. (2024) uses MoE across the spatial domain with non-overlapping patches to decompose global physical hard constraints into multiple local constraints. Our PoU-MoE trunk uses a similar methodology where it has individual expert trunk networks on each patch in the domain, albeit with overlapping patches. However, instead of learning physical constraints, we let our experts learn *spatially local* basis functions (see Appendix G for further discussion on this spatial locality).

## B    Speculation on an ensemble FNO

Here, we show one possible technique for incorporating the PoU-MoE localized bases into the FNO architecture, *i.e.*, we show how to create an ensemble FNO. FNOs consist of a *lifting* operator that lifts the input functions to multiple channels, a *projection* operator that undoes the lift, and intermediate layers (Fourier layers) consisting of kernel-based integral operators discretized by the fast Fourier transform (FFT); these integral operators are also typically augmented by pointwise convolution operations. Let $f_t$ denote the intermediate function at the $t^{th}$ Fourier layer. Then, the output $f_{t+1}$ of this layer (and the input to the next layer) is given by

$$f_{t+1}(y) = \sigma \left( \int_{\Omega} \mathcal{K}(x,y) f_t(x) \ dx \ + \ W f_t(y) \right), \ x \in \Omega, \tag{15}$$

where $\sigma$ is an activation function applied pointwise, $\mathcal{K}$ is a matrix-valued kernel learned in Fourier space via the FFT, and $W$ is the aforementioned pointwise convolution (Li et al., 2021). Since FNOs use the FFT to compute the integral operator in (15), this effectively constitutes a projection of $f_t(x)$ onto a set of *global* Fourier modes (trigonometric polynomials or complex exponentials). One possible method for creating an ensemble FNO would involve modifying (15) to incorporate a set of localized basis functions using the PoU-MoE formulation as follows:

$$f_{t+1}(y) = \sigma \left( \underbrace{\int_{\Omega} \mathcal{K}(x,y) f_t(x) \ dx}_{\text{Global basis}} + \underbrace{\sum_{k=1}^{P} w_k(y) \int_{\Omega_k} \mathcal{K}(x,y) \ f_t(x)|_{\Omega_k} \ dx}_{\text{Localized basis}} + W f_t(y) \right), \tag{16}$$

where $P$ is the number of spatial patches (all of which are hypercubes). The PoU-MoE formulation now combines a set of *localized* integrals on each patch, each of which when computed by an FFT would constitute a projection of $f_t$ (restricted to $\Omega_k$) onto a local Fourier basis. This loosely resembles the Chebyshev polynomial PoU approximation introduced by Aiton & Driscoll (2018).

It is worth mentioning that this is one of many ways to combine different basis functions in FNOs. Another way is to introduce a set of local basis functions at the final projection operator that maps to

the output function. The projection operator's final layer can be enlarged to weight the additional basis functions, closely resembling how the branch weights the ensemble trunk in ensemble DeepONets. Similar extensions are possible for the GNO and even kernel/GP-based operator learning techniques.

## C    SUBOPTIMAL ENSEMBLE TRUNK ARCHITECTURES

We document here our experience with other ensemble trunk architectures. We primarily made the following two other attempts:

**A residual ensemble**: Our first attempt was to combine the different trunk outputs using weighted residual connections with trainable weights, then activate the resulting output, then pass that activated output to a dense layer. For instance, given two trunks $\boldsymbol{\tau}_1$ and $\boldsymbol{\tau}_2$, this residual ensemble trunk would be given by

$$\hat{\boldsymbol{\tau}}_{\text{res}} = W\sigma\left(\tanh(w_1)\boldsymbol{\tau}_1 + \tanh(w_2)\boldsymbol{\tau}_2\right) + b, \tag{17}$$

where $\sigma$ was some nonlinear activation, $W$ was some matrix of weights, and $b$ a bias. We also attempted using the sigmoid instead of the tanh. The major drawback of this architecture was that the output dimensions of the individual trunks had to match, *i.e.*, $p_1 = p_2$ to add the results (otherwise, some form of padding would be needed). We found that this architecture indeed outperformed the vanilla-DeepONet in some of our test cases, but required greater fine tuning of the output dimension $p$. In addition, we found that this residual ensemble failed to match the accuracy of our final ensemble architecture.

**An activated ensemble**: Our second attempt resembled our final architecture, but had an extra activation function and weights and biases. This activated ensemble trunk would be given by

$$\hat{\boldsymbol{\tau}}_{\text{act}} = W\sigma\left([\tau_1, \tau_2]\right) + b. \tag{18}$$

This architecture allowed for different $p$ dimensions (columns) in $\tau_1$ and $\tau_2$. However, we found that this architecture did not perform well when the POD trunk was one of the constituents of the ensemble; this is likely because it is suboptimal to activate a POD trunk, which is already a data-dependent basis. There would also be no point in moving the activation function onto the other ensemble trunk constituents, since these are always activated if they are not POD trunks. Finally, though $W$ and $b$ allowed for a trainable combination rather than simple stacking, they did not offer greater expressivity over simply allowing a wider branch to combine these different trunks. We found that this architecture also underperformed our final reported architecture.

## D    PROOF OF UNIVERSAL APPROXIMATION THEOREM FOR THE POU-MOE DEEPONET

We have

$$\|\mathcal{G}(u)(y) - \mathcal{G}^\dagger(u)(y)\|_{\mathcal{V}} = \left\|\mathcal{G}(u)(y) - \left\langle \boldsymbol{\beta}(u;\theta_b), \sum_{j=1}^{P} w_j(y)\boldsymbol{\tau}_j(y;\theta_{\boldsymbol{\tau}_j})\right\rangle - b_0\right\|_{\mathcal{V}},$$

$$= \left\|\underbrace{\left(\sum_{j=1}^{P} w_j(y)\right)}_{=1}\mathcal{G}(u)(y) - \left\langle \boldsymbol{\beta}(u;\theta_b), \sum_{j=1}^{P} w_j(y)\boldsymbol{\tau}_j(y;\theta_{\boldsymbol{\tau}_j})\right\rangle\right.$$

$$\left. - \underbrace{\left(\sum_{j=1}^{P} w_j(y)\right)}_{=1} b_0\right\|_{\mathcal{V}},$$

$$= \left\|\sum_{j=1}^{P} w_j(y)\left(G(u)(y) - \left\langle \boldsymbol{\beta}(u;\theta_b), \boldsymbol{\tau}_j(y;\theta_{\boldsymbol{\tau}_j})\right\rangle - b_0\right)\right\|_{\mathcal{V}},$$

$$\leq \sum_{j=1}^{P} w_j(y)\|\mathcal{G}(u)(y) - \left\langle \boldsymbol{\beta}(u;\theta_b), \boldsymbol{\tau}_j(y;\theta_{\boldsymbol{\tau}_j})\right\rangle - b_0\|_{\mathcal{V}}.$$

Given a branch network $\boldsymbol{\beta}$ that can approximate functionals to arbitrary accuracy, the (generalized) universal approximation theorem for operators automatically implies that (Chen & Chen, 1995; Lu et al., 2021) a trunk network $\boldsymbol{\tau}_j$ (given sufficient capacity and proper training) can approximate the restriction of $\mathcal{G}$ to the support of $w_i(\mathbf{y})$ such that:

$$\|\mathcal{G}(u)(y) - \langle \boldsymbol{\beta}(u; \theta_b), \boldsymbol{\tau}_j(y; \theta_{\boldsymbol{\tau}_j})\rangle - b_0\|_{\mathcal{V}} \le \epsilon_j,$$

for all $y$ in the support of $w_j$ and any $\epsilon_j > 0$. Setting $\epsilon_j = \epsilon$, $j = 1, \ldots, P$, we obtain:

$$\|\mathcal{G}(u)(y) - \mathcal{G}^\dagger(u)(y)\|_{\mathcal{V}} \le \epsilon \underbrace{\sum_{j=1}^{P} w_i(y)}_{=1},$$

$$\implies \|\mathcal{G}(u)(y) - \mathcal{G}^\dagger(u)(y)\|_{\mathcal{V}} \le \epsilon.$$

where $\epsilon > 0$ can be made arbitrarily small. This completes the proof.

## E  HYPERPARAMETERS

### E.1  NETWORK ARCHITECTURE

In this section, we describe the architecture details of branch and trunk networks. The architecture type, size, and activation functions are listed in Table 5. The CNN architecture consists of two five-filter convolutional layers with 64 and 128 channels respectively, followed by a linear layer with 128 nodes. Following Lu et al. (2021), the last layer in the branch network does not use an activation function, while the last layer in the trunk does. The individual PoU-MoE trunks in the ensemble models also use the same architecture as the vanilla trunk. We use the *unstacked* DeepONet with bias everywhere (except the POD-DeepONet which does not use a bias).

Table 5: DeepONet network architectures across all models and problems. The CNN architecture is described in Appendix E.1.

|                       | Branch              | Trunk                  | Activation function |
| --------------------- | ------------------- | ---------------------- | ------------------- |
| Darcy flow            | 3 layers, 128 nodes | 3 layers, 64 nodes     | Leaky-ReLU          |
| 2D Reaction-Diffusion | CNN                 | 3 layers, 128 nodes    | ReLU                |
| Cavity flow           | CNN                 | [128, 128, 128, 100]   | tanh                |
| 3D Reaction-Diffusion | 3 layers, 128 nodes | 3 layers, 128 nodes    | ReLU                |

### E.2  OUTPUT DIMENSION $p$

We list the relevant DeepONet hyperparameters we use below. The $p$ ($p_{\text{POD}}$ for POD) values are listed in Table 6 for all the DeepONets.

Table 6: $p$ ($p_{\text{POD}}$ for POD) values for the various DeepONet models. For $(P + 1)$-vanilla DeepONet, the total number of basis functions is shown below. RD stands for reaction-diffusion.

|                  | Darcy flow      | Cavity flow   | 2D RD           | 3D RD           |
| ---------------- | --------------- | ------------- | --------------- | --------------- |
| Vanilla          | 100             | 100           | 100             | 100             |
| POD              | 20              | 6             | 20              | 20              |
| Modified-POD     | 20              | 6             | 20              | 20              |
| (Vanilla, POD)   | (100, 20)       | (100, 6)      | (100, 20)       | (100, 20)       |
| $(P + 1)$-Vanilla | 400            | 500           | 700             | 900             |
| Vanilla-PoU      | (100, 100)      | (100, 100)    | (100, 100)      | (100, 100)      |
| POD-PoU          | (20, 100)       | (6, 100)      | (20, 100)       | (20, 100)       |
| Vanilla-POD-PoU  | (100, 20, 100)  | (100, 6, 100) | (100, 20, 100)  | (100, 20, 100)  |

### E.3 PARTITIONING

The PoU-MoE trunk has certain hyperparameters that must be chosen. In our experiments, to maximize accuracy, we chose the patch size and the number of patches that produced the smallest possible patches and the smallest number of patches, while simultaneously seeking that the domain was covered and ensuring that the patches did not extend too far outside the domain boundary. Coincidentally, this strategy coincided with placing individual patches over regions of high spatial error in the vanilla-DeepONet solution (effectively, patches over "features of interest"). In the reaction-diffusion examples, even though we used uniform patch radii, we ensured that the patches did not overlap horizontally over line of the discontinuity. This choice combined with the use of ReLU activation ensured that we resolved that discontinuity better than vanilla-DeepONet; we believe this is one of the unique strengths of the PoU-MoE approach. Currently, we use the same trunk architectures on each patch as in the vanilla-DeepONet. In future work, adaptive patch selection strategies (such as making the patch centers and radii trainable or enforcing soft constraints on them as part of the loss function) can be used to automate determining patch placement and patch size. Furthermore, the patch shape can be changed depending on the problem domain; elongated/ellipsoidal patches can be used in narrower regions where spherical patches are not well suited.

## F ADDITIONAL RESULTS

We present additional results and figures in this section related to the problems in Section 3.

### F.1 2D DARCY FLOW

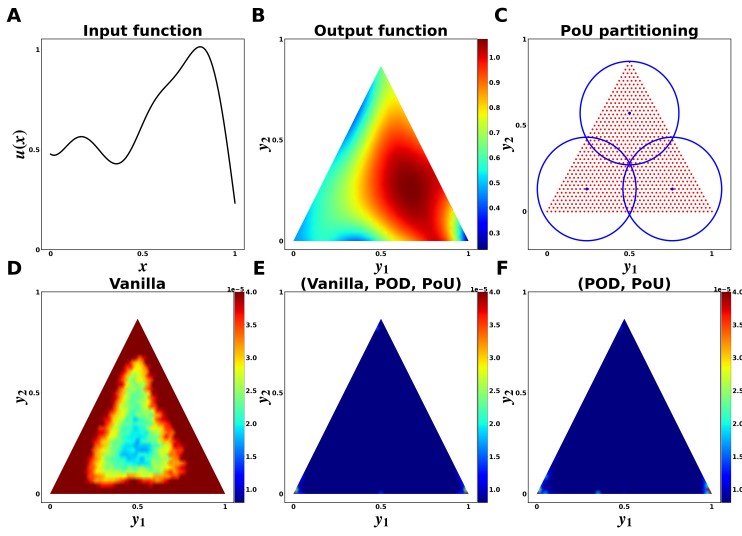

Figure 6: The 2D **Darcy flow** problem. (**A**) and (**B**) show example input and output functions respectively. (**C**) shows the three patches used for the PoU-MoE trunk. (**D**), (**E**), and (**F**) show the spatial mean squared error (MSE) for the vanilla, ensemble vanilla-POD-PoU, and ensemble POD-PoU DeepONets respectively.

The 2D Darcy flow problem models fluid flow within a porous media. The flow's pressure field $u(y)$ and the boundary condition are given by

$$-\nabla \cdot (K(y) \nabla u(y)) = f(y), \; y \in \Omega, \tag{19}$$

$$u(y) \sim \mathcal{GP}\left(0, \mathcal{K}(y_1, y_1')\right), \tag{20}$$

where $K(y)$ is the permeability field, and $f(y)$ is the forcing term. The Dirichlet boundary condition was sampled from a zero-mean Gaussian process with a Gaussian kernel as the covariance function; the kernel length scale was $\sigma = 0.2$. As in Lu et al. (2022), we learned the operator $\mathcal{G} : u(y)|_{\partial\Omega} \to u(y)|_{\Omega}$. We used the dataset provided in Lu et al. (2022) which contains 1900 training and 100 test input and output function pairs. $\Omega$ was a triangular domain (shown in Figure 6). The permeability field and the forcing term were set to $K(y) = 0.1$ and $f(y) = -1$. Example input and output functions, and the three patches for PoU trunks are shown in Figure 6. The partitioning always

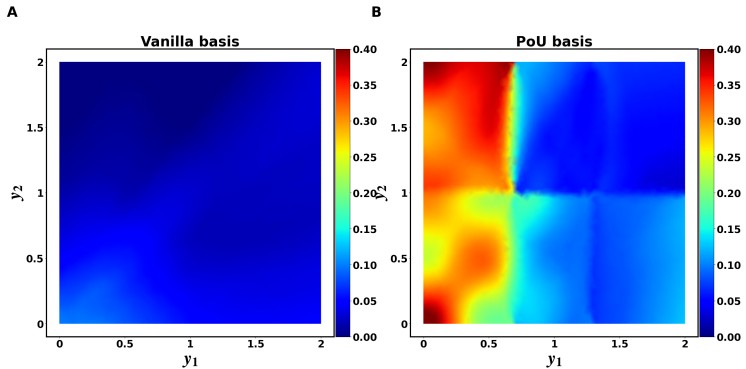

Figure 7: Vanilla-DeepONet and PoU (from POD-PoU ensemble DeepONet) basis functions for the largest branch modes on the 2D **reaction-diffusion** problem.

ensures that the regions with high spatial gradients are captured completely or near-completely by a patch.

We report the relative $\ell_2$ errors (as percentages) on the test dataset for the all the models in Table 1. The vanilla-POD-PoU ensemble was the most accurate model with a 4.5x error reduction over the vanilla-DeepONet and a 1.5x reduction over our POD-DeepOnet. The POD-PoU ensemble was second best with a 3.7x error reduction over the vanilla-DeepONet and a 1.5x reduction over the POD-DeepONet. The highly overparametrized $(P + 1)$-vanilla model was **less accurate** than the standalone DeepONets. On this problem, overparametrization appeared to help only when spatial localization was also present; the biggest impact appeared to be from having both the right global and local information. The MSE errors as shown in Figure 6 corroborate these findings.

## G    EVIDENCE FOR SPATIAL LOCALIZATION OF THE POU-MOE BASIS

Here, we present further evidence showing that the PoU-MoE trunk learns spatially local features. In Figure 7, we show basis functions from the vanilla-DeepONet trunk and the PoU-MoE trunk of the POD-PoU ensemble DeepONet. Unlike the basis functions shown in Figure 2, these correspond to the largest branch coefficients in the respective models, i.e., the most "important" basis functions. Clearly, the PoU basis has a significantly higher spatial variation than the vanilla basis. We believe that this learned *spatial locality* helps the ensemble DeepONets with the PoU-MoE trunk achieve superior accuracy on problems with strong local features (such as those tested in this work).

