# OpenReview forum: "Ensemble and Mixture-of-Experts DeepONets For Operator Learning"
_ICLR.cc/2025/Conference — ICLR 2025 Conference Withdrawn Submission_

### Official Review · Reviewer_diRF · 2024-10-22

**Soundness:** 3
**Presentation:** 3
**Contribution:** 3
**Rating:** 5
**Confidence:** 4

**Summary:**

This paper presents a spatial mixture-of-experts (MoE) DeepONet architecture that utilizes partition-of-unity (PoU) approximation to combine expert networks across overlapping spatial patches. By integrating this localized approach with proper orthogonal decomposition (POD), the authors achieve 2-4x lower errors compared to standard DeepONets on several PDE problems. Besides, the authors also provide theoretical guarantees  for the proposed architecture through universal approximation theorems.

**Strengths:**

1. The paper presents its technical content in a clear, organized manner.

2. The integration of partition-of-unity principles into the DeepONet framework represents an interesting approach to incorporating spatial locality.

3. The work combines theoretical analysis (universal approximation theorems) with systematic empirical validation.

4. The proposed method demonstrates consistent performance improvements over standard DeepONets across multiple PDE examples.

**Weaknesses:**

1.  The experimental comparisons are primarily focused on DeepONet variants, omitting comparisons with other popular neural operators like FNO, which would provide broader context for the method's effectiveness.

2. The time-dependent PDE examples are restricted to single-step predictions (from one time point to another), leaving open questions about the method's capability to learn full temporal trajectories when time coordinates are included in the trunk network inputs.

3. The dependence on predefined partitions may limit the method's flexibility and generalizability, particularly for problems where optimal partition locations are not known a priori.

**Questions:**

1. How sensitive is the method's performance to the number of partitions? It would be valuable to see an ablation study on this hyperparameter.

2. Given that POD bases are computed from discretized output functions, how does the method handle evaluation at arbitrary points y not present in the training discretization?

3. How well does the method generalize to learning mappings from initial conditions to full spatiotemporal solutions, rather than just single-time predictions?

---

### Official Review · Reviewer_okGg · 2024-10-28

**Soundness:** 3
**Presentation:** 2
**Contribution:** 2
**Rating:** 3
**Confidence:** 3

**Summary:**

This paper introduces an innovative enhancement for DeepONets, integrating multiple trunk networks to boost expressivity and generalization. It proposes a partition-of-unity mixture-of-experts (PoU-MoE) trunk structure to promote spatial locality, offering a refined approach to operator learning. Theoretical guarantees and extensive experiments across various PDE problems reveal that ensemble DeepONets, particularly the POD-PoU variant, achieve error reductions of 2-4x compared to standard DeepONets. This work offers valuable insights into effective trunk configurations and highlights a promising direction for advancing operator learning, albeit with increased computational requirements.

**Strengths:**

- The partition-of-unity mixture-of-experts (PoU-MoE) trunk introduces a novel approach that enhances spatial locality and promotes model sparsity.

- As reported, the ensemble DeepONets, particularly the POD-PoU variant, achieve substantial error reductions (2-4x) compared to standard DeepONets.

- Universal approximation capabilities are analyzed.

**Weaknesses:**

- The presentation could be improved. For instance, a clear description and detailed experimental setup for each baseline in Table 1 should be provided.
- The scope of this work appears somewhat limited, as it primarily focuses on testing enrichment strategies for basis functions within the specific context of operator learning. Applying these strategies to other popular frameworks is not trivial. Although the authors suggest that these methods might extend to FNO, sufficient details and evidence to substantiate this claim are lacking (Appendix B is not convincing enough).

- No comparison with other popular frameworks e.g. CNO [1], FNO, variants of FNO, etc.. Although this is also partially due to the fact that the scope of this work is limited to DeepONet.

- While DeepONet is among the most well-known neural operators with clearly identifiable basis functions (the trunks), exploring additional works could help assess the generalizability of the proposed frameworks, such as [2] (treating the INRs/SIREN as producing basis functions as in DeepONet) .

[1] Convolutional Neural Operators for robust and accurate learning of PDEs; Bogdan Raonić, Roberto Molinaro, Tim De Ryck, Tobias Rohner, Francesca Bartolucci, Rima Alaifari, Siddhartha Mishra, Emmanuel de Bézenac; 2023

[2] Operator Learning with Neural Fields: Tackling PDEs on General Geometries; Louis Serrano, Lise Le Boudec, Armand Kassaï Koupaï, Thomas X Wang, Yuan Yin, Jean-Noël Vittaut, Patrick Gallinari; 2024

**Questions:**

1. Can you clearly explain what each of these baselines are in Table 1?
2. The choice of testing datasets are not so conventional, how does it perform on e.g. NS from Li et. 2020?
3. Why is (P + 1)-Vanilla MUCH worse than Vanilla (ours)? To my understanding, you simply add a layer to (P + 1)? Then with residual connection, it should be at least similar to Vanilla? Is there no residuals in your network?

---

### Official Review · Reviewer_iEnf · 2024-11-02

**Soundness:** 3
**Presentation:** 3
**Contribution:** 2
**Rating:** 5
**Confidence:** 4

**Summary:**

This paper applies classical mixture-of-expert paradigm to learn mathematical operators. By incorporating different experts, and therefore enhancing basis representation, the resulting network has stronger learning power.

**Strengths:**

1. Integrating the mixture-of-experts paradigm into operator learning enhances the model's capacity for effectively learning operators.
2. Building on the mixture-of-experts approach, the authors introduce a partition-of-utility strategy to encourage spatial locality and promote model sparsity.
3. A MoE model enhances the model's capacity by incorporating a diverse set of basis functions, which enables improved approximation accuracy across various operator learning tasks.
4. Comprehensive ablation study regarding key design factors of the MoE framework.

**Weaknesses:**

While the paper claims to present a novel framework combining classical expert neural networks, it largely repackages existing concepts rather than introducing groundbreaking ideas. The mixture-of-experts (MoE) paradigm is a well-established approach in machine learning, although innovative when applied in scientific ML, does not fundamentally transform the field. The combination of these elements lacks a compelling new problem formulation or a significant shift in methodology. The quality of the paper is undermined by several factors. There is a lack of critical evaluation of the scenarios where the QE-MoE approach may fail or perform suboptimally, especially concerning the boundary learning. While distributing data spatially to different expert models could potential relief the learning complexity compared to learning the data globally, it also introduces significant complexities depending on number of mixtures and data partitioning strategy. The experimental results do not convincingly demonstrate a substantial improvement over state-of-the-art methods but a comparison over several of its MoE variants.

**Questions:**

See above.

**Details Of Ethics Concerns:**

no.

---

### Note · Authors · 2024-11-13

I have read and agree with the venue's withdrawal policy on behalf of myself and my co-authors.